# Stressing State Analysis of SRC Column with Modeling Test and Finite Element Model Data

**Zijie Shen, Bai Liu * and Guangchun Zhou**

Key Lab of Smart Prevention and Mitigation of Civil Engineering Disasters of the Ministry of Industry and Information Technology, School of Civil Engineering, Harbin Institute of Technology, Harbin 150090, China

\* Correspondence: liubai_12138@163.com; Tel.: +86-15735515958

**Abstract:** This paper reveals the failure characteristic points of the spiral reinforced column during the damage process by modeling and analyzing the stressing state of the column with the test and finite element output data. At the same time, the structural stressing state theory and the correlation modeling analysis method's applicability to spiral reinforced concrete columns are verified. First, a finite element model was established based on the literature's spiral reinforced concrete column tests. Then, correlation modeling was performed on the test strain data to obtain correlation characteristic pairs (mode-characteristic parameters), and stressing state modeling was performed on the internal energy and element strain energy data from the finite element model to obtain stressing state characteristic pairs. The slope increment criterion is applied to the obtained stressing state characteristic parameter curves to reveal the characteristic point Q, defined as the failure starting point. The reasonableness of the failure starting point is further verified by observing the cloud diagram of the finite element model in the vicinity of the characteristic point Q. In general, the correlation modeling method proposed in this paper can provide a new reference for structural stressing state analysis. In addition, the failure starting point of spiral reinforced concrete columns revealed in this paper can be used as a design reference.

**Keywords:** SRC column; finite element model; correlation modeling method; structural stressing state theory; characteristic pair; characteristic points

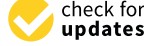



## 1. Introduction

Reinforced concrete structures are widely used in various engineering structures all over the world. Spiral reinforcement concrete (SRC) columns have apparent nonlinear behavior during the loading process due to the restraint effect of stirrups on the core concrete. Therefore, it is very typical among reinforced concrete structures. Moreover, SRC columns are widely used in engineering due to their high bearing capacity, deformation capacity, and energy consumption capacity [1,2]. Since the beginning of the twentieth century, many scholars worldwide have studied the bearing capacity of SRC columns from the perspective of constitutive models of confined concrete and concrete strength criteria [3,4].

Taking the Chinese code for concrete structures GB 50010-2010 [5] as an example, three contributions are considered when calculating the axial bearing capacity of SRC columns: core concrete, vertical bar, and stirrups' restraint effect. Moreover, the three parts' contributions to the SRC column's bearing capacity are linearly superimposed, as shown in Equation (1):

$$N \le 0.9(f_c A_{\mathrm{cor}} + f_y' A_s' + 2\alpha f_{\mathrm{yv}} A_{\mathrm{ss0}}) \tag{1}$$

$$A_{\mathrm{ss0}} = \frac{\pi d_{\mathrm{cor}} A_{\mathrm{ssl}}}{s} \tag{2}$$

Most of the world's concrete structure codes [5–7] have similar ideas when calculating the ultimate bearing capacity of SRC columns or design calculations. Therefore, the prob-

lem of calculating the ultimate bearing capacity of SRC columns is transformed into the process of establishing the constitutive relationship of concrete confined by stirrups and determining the peak stress/strain. There are many classic and well-known constrained concrete constitutive models, including the Sheikh model [1], Mander model [8], Saatcioglu model [9], and Hoshikuma model [10], etc. Tables 1 and 2 are the peak stress/strain improvement coefficients of the classic confined concrete constitutive model and the expressions of the rising section, falling section, and softening coefficient of the constitutive model.

**Table 1.** Peak stress/strain factor of SRC column.

| Classical Constitutive Model | $K_s = \frac{f_{cc}}{f_{co}}$ | $K_d = \frac{\varepsilon_{cc}}{\varepsilon_{co}}$ |
|---|---|---|
| Sheikh SA, Uzumeri SM [1] | $1 + \frac{b_c^2}{140 P_{occ}} \left[ \left(1 - \frac{nC^2}{5.5 b_c^2}\right)\left(1 - \frac{s}{2b_c}\right)^2 \right]$ | $1 + \frac{248}{C}\left[1 - 5.0\left(\frac{s}{B}\right)^2\right]\frac{\rho_{sh} f_s'}{\sqrt{f_c'}}$ |
| Mander et al. [8] | $-1.254 + 2.254\sqrt{1 + 7.94\frac{f_l'}{f_{co}}} - 2\frac{f_l'}{f_{co}}$ | $1 + 5\left[\frac{f_{cc}'}{f_{co}} - 1\right]$ |
| Saatcioglu and Razvi [9] | $1 + k_1 \frac{f_{le}}{f_{co}}$ | $1 + 5k_1 \frac{f_{le}}{f_{co}}$ |
| Hoshikuma et al. [10] | $1 + 0.73\frac{\rho_{sh} f_{yh}}{f_{co}}$ | $1 + 4.98\frac{\rho_{sh} f_{yh}}{f_{co}}$ |

**Table 2.** Expression of constitutive model of SRC column.

| Classical Constitutive Model | Ascending Segment | Descending Segment | Softening Factor |
|---|---|---|---|
| Sheikh SA, Uzumeri SM [1] | $f_c = f_{cc}\left[2x - x^2\right]$ | $f_c = f_{cc} - Z(\varepsilon_c - \varepsilon_{c2}) \geq 0.3 f_{cc}$ | $Z = \frac{0.5}{\frac{3}{4}\rho_{sh}\sqrt{\frac{B}{s}}}$ |
| Mander et al. [8] | $f_c = \frac{f_{cc}mr}{r - 1 + m^r}$ | | |
| Saatcioglu and Razvi [9] | $f_c = f_{cc}\left[2x - x^2\right]$ | $f_c = f_{cc} - Z(\varepsilon_c - \varepsilon_{cc}) \geq 0.2 f_{cc}$ | $Z = \frac{0.15 f_{cc}}{\varepsilon_{85} - \varepsilon_{cc}}$ |
| Hoshikuma et al. [10] | $f_c = E_c \varepsilon_c \left[1 - \frac{1}{n} x^{n-1}\right]$ | $f_c = f_{cc} - E_{des}(\varepsilon_c - \varepsilon_{cc}) \geq 0.5 f_{cc}$ | $E_{des} = \frac{11.2 f_{co}^2}{\rho_{sh} f_{yh}}$ |

It can be seen from Tables 1 and 2 that even for the classical confined concrete constitutive model, there is a huge difference in form between the peak stress/strain and the expression of the constitutive model. In addition, the constitutive model has a more significant gap in the mechanism. For example, the Mander model does not consider the softening coefficient in the expression; for the Saatcioglu constitutive model, the softening coefficient is mainly affected by the peak stress/strain: For the Sheikh constitutive model, the influencing factor of the softening coefficient is the structure of the SRC column.

The fundamental reason for the considerable difference in the form of these expressions is that the working behavior of concrete is difficult to describe accurately; the ultimate state of concrete has substantial uncertainties. There are still many studies on the ultimate bearing capacity of SRC columns based on the constitutive model of concrete confined by stirrups and the ultimate state. For example, Nagashima [11], Issa MA [12], Suzuki M [13], Akiyama M [14], Wu T [15], and others proposed various constitutive models and peak stresses of concrete confined by stirrups from 1992 to 2018. These models and coefficients can often only better predict the working behavior of partially constructed stirrup-constrained concrete. Therefore, the peak bearing capacity of SRC columns was obtained based on these constitutive models, and the peak stress/strain improvement factor has the characteristics of substantial uncertainty. In addition, there is relatively little research on SRC columns based on numerical simulation, but the essence of numerical simulation from concrete structures is the selection of constitutive models and the determination of ultimate states. Therefore, the peak bearing capacity of SRC columns was obtained based on these constitutive models' uncertainty. In general, the study of SRC column tests and finite element studies has the following problems:

(1) The load-carrying capacity of SRC columns obtained based on the above restrained concrete principal structure model has significant uncertainty;

(2)  In engineering or research, force-displacement curves are often used to describe the evolution of SRC columns' working behavior and determine their load-carrying capacity. The strain or strain energy data output from tests or finite element models is not well utilized;

(3)  The SRC column bearing capacity study did not consider the coordinated working performance between the hoop reinforcement and the core concrete well.

The above problems are common in structural engineering research. Zhou [16–26] deeply considered the above problems and challenged the uncertainty in the structural analysis process. Zhou emphasized that structural failure is an evolutionary process with a definite starting point, and there are apparent mutation laws and interaction laws between elements in the evolutionary process. The failure starting point of a structure, i.e., the moment when the stressing state of a structure changes from a steady state to an unsteady state, is more valuable for structural analysis and design. The general damage laws revealed based on the damage starting point of a structure have been: statically loaded steel nodes [17]; hysterically loaded reinforced masonry shear walls [18,19]; steel frames [20]; steel pipe concrete columns [21]; concrete filled steel pipe arches [22,23]; concrete filled steel pipe arch bridges [24]; continuous curved box girder bridges [25]; space mesh shells [26] and other structures; more than a dozen structures have been verified under different working conditions.

In this paper, the axial loading test data of SRC columns from the literature [27] were cited, and finite element models were developed. Then, based on the structural stressing state theory and correlation modeling analysis method, the test and simulation data of SRC columns are modeled as stressing state characteristic pairs (mode-characteristic parameters). Next, the characteristic points P with deterministic characteristics are obtained by mutation analysis of the stressing state characteristic pairs and defined as the failure starting point. The reasonableness of the failure threshold is demonstrated by comparing the failure threshold obtained from the modeling tests and simulations with the results of the cloud diagram analysis.

## 2. Modeling Test/Simulation Data

### 2.1. Overview of Stressing State Theory

Traditional structural analysis often revolves around the intrinsic structure model, ultimate states, damage modes, and peak load carrying capacity. However, the structural stressing state theory proposed by Zhou [16–26] argues that by modeling the displacement, strain, stress, and strain energy of the test or finite element method as the stressing state characteristic pairs (mode-characteristic parameters) can characterize the structural stressing state and reveal its failure characteristics. Moreover, the obtained characteristic points are defined as the starting point of failure, which is a very stable characteristic point and can be used as a design reference point for the structure. The process of stressing state analysis based on test/simulation data in this paper is shown in Figure 1.

### 2.2. Modeling Internal Energy and Element Strain Energy

This paper characterizes the evolution of the stressing state of the SRC column finite element model (EFM) based on the internal energy (*IE*) input to the whole structure and concrete, stirrups and vertical bars during the loading process, as shown in Equation (3):

$$IE_{j,\text{norm}} = \frac{IE_j^c + IE_j^v + IE_j^s}{Max(IE_j^c + IE_j^v + IE_j^s)} \tag{3}$$

$$\Delta IE_{j,\text{norm}} = \frac{(IE_{j+1} - IE_j)}{Max(IE_{j+1} - IE_j)} \tag{4}$$

$$\Delta^2 IE_{j,\text{norm}} = \frac{(IE_{j+1} - 2IE_j + IE_{j-1})}{Max(IE_{j+1} - 2IE_j + IE_{j-1})} \tag{5}$$

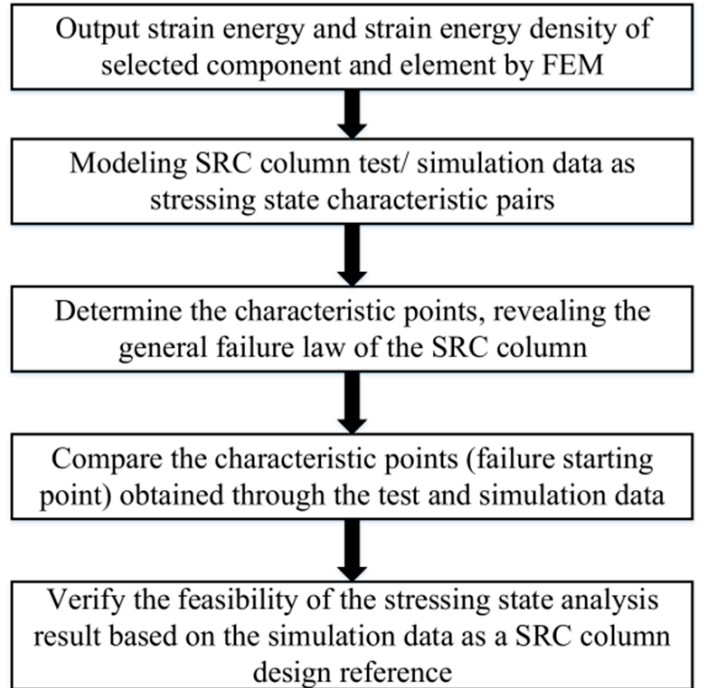

**Figure 1.** Process of SRC column's stressing state analysis.

*IE* represents the internal energy of each part output by the post-processing module of the SRC column's finite element model. The superscripts c, v, and s represent the concrete, vertical steel bar, and spiral steel bar of the SRC column, respectively. The subscript j represents the load step in the process of FEM analysis. As shown in Equations (4) and (5), in order to further reveal the mutation law in the process of stressing state evolution of SRC columns, the first-order characteristic parameters $\Delta IE_{j,\text{norm}}$ and second-order characteristic parameters $\Delta^2 IE_{j,\text{norm}}$ are also defined in this paper to characterize the stressing state evolution of SRC columns.

In this paper, 10 elements in the concrete part of the SRC column finite element model, 14 elements in the high-strength stirrup part, and 12 elements in the longitudinal steel bar are selected to establish the stressing state sub-modes. Based on the element strain energy output by the ABAQUS post-processing module, a sub-mode of the stressing state is established as shown in Equation (6):

$$\mathbf{S}_j^c = [ESE_1^c, ESE_2^c, \dots, ESE_{10}^c], \ \mathbf{S}_j^v = [ESE_1^v, ESE_2^v, \dots, ESE_{12}^v], \ \mathbf{S}_j^s = [ESE_1^s, ESE_2^s, \dots, ESE_{14}^s] \quad (6)$$

In order to eliminate the influence of the numerical value of the element strain energy, the evolution law of the stressing state of the structure can be more directly characterized. Normalize the element strain energy (*ESE*) of the sub-mode to obtain the normalized stressing state sub-mode. The expression of normalized *ESE* is shown in Equation (7):

$$ESE_{j,\text{norm}} = \frac{ESE_j}{ESE_M} \quad (7)$$

The subscript *M* represents the maximum value of *ESE* when the SRC column finite element model is loaded from 0 to the ultimate bearing capacity in axial loading. By further modeling the element strain energy of the SRC column, the corresponding characteristic parameters can be obtained.

### 2.3. Correlation Modeling Method

The correlation modeling method proposed in this paper for the stressing state analysis of SRC columns treats the studied structure or member as a whole and characterizes the evolution of the stressing state of the structure or member through the interaction between the modeled elements. The characteristic pairs established based on the correlation method are called correlation characteristic pairs. The correlation modeling analysis method is introduced below as an example of modeling SRC column test data.

Step 1: Select 4 measurement points on the surface of concrete or spiral reinforcement to output the strain data of the whole process of loading, integrate it against the load and take the absolute value. The matrix shown as Equation (8) is obtained:

$$\mathbf{S}_j^{\text{Step1}} = \left[ \left| \int_0^F \varepsilon_1 \mathrm{d}F \right| \cdots \left| \int_0^F \varepsilon_4 \mathrm{d}F \right| \right]_j^4 \tag{8}$$

where $\varepsilon$ denotes the strain in concrete or spiral reinforcement, the subscript $j$ denotes the ordinal number of load steps, $F$ denotes the load step in the loading process, and the number 4 in the superscript denotes the number of elements in the vector. The symbols in the following formula have the same meaning. Then the elements in the Equation (8) vector are taken two by two to obtain the vector shown in Equation (9):

$$\mathbf{S}_j^{\text{Difference1}} = \left[ \left| \int_0^F \varepsilon_1 \mathrm{d}F \right| - \left| \int_0^F \varepsilon_2 \mathrm{d}F \right| \cdots \left| \int_0^F \varepsilon_3 \mathrm{d}F \right| - \left| \int_0^F \varepsilon_4 \mathrm{d}F \right| \right]_j^6 \tag{9}$$

$$\mathbf{S}_j^{\text{Difference1}} = [D_{1-2} \cdots D_{3-4}]_j^6 = \left[ D_a \cdots D_f \right]_j^6 \tag{10}$$

The vector Equation (9) is simplified to obtain the vector shown in Equation (10), and the superscript 6 indicates the number of elements in the vector.

Step 2: Integrate the elements in the vector Equation (10), take the absolute value and then normalize to get the vector shown in Equation (11):

$$\mathbf{R}_{j,\text{norm}}^{\text{Step2}} = \left[ \left| \int_0^F D_a \mathrm{d}F \right| \cdots \left| \int_0^F D_f \mathrm{d}F \right| \right]_{j,\text{norm}}^6 \tag{11}$$

$$\mathbf{R}_j^{\text{sub}} = \left[ \left( \left| \int_0^F D_a \mathrm{d}F \right|_{\text{norm}} - \left| \int_0^F D_b \mathrm{d}F \right|_{\text{norm}} \right) \cdots \left( \left| \int_0^F D_a \mathrm{d}F \right|_{\text{norm}} - \left| \int_0^F D_b \mathrm{d}F \right|_{\text{norm}} \right) \right]_j^{15} \tag{12}$$

$$\mathbf{R}_j^{\text{sub}} = \left[ E_{ab} \cdots E_{ef} \right]_j^{15} \tag{13}$$

The vector shown in Equation (12) is obtained by taking the difference in the elements in the vector of Equation (11) two by two. Superscript 15 indicates the number of elements in the vector. The expression of Equation (12) is simplified as shown in Equation (13). Integrate the elements in Equation (13), take the absolute values, and then sum up and normalize to obtain the correlation characteristic parameter shown in Equation (14):

$$R_{j,\text{norm}}^{\text{sub}} = \left( \sum_{N=1}^{15} \left| \int_0^F E_{ab} \mathrm{d}F \right| \right)_{j,\text{norm}} \tag{14}$$

The correlation modeling method for the output data of the SRC column finite element model is similar to the above steps and will not be repeated here.

### 2.4. Slope Increment Criterion

The evolution of the structural stressing state is a continuous process. The point where the mutation of the characteristic parameter curve occurs is defined as the characteristic point P. Where the screening of the mutation point is based on the slope increment criterion. The expression is shown in Equation (15):

$$S_j - S_{j-1} \geq W \tag{15}$$

Among them, *S* represents the slope of the characteristic parameter curve, *j* represents the load step, and *W* is the limit determined based on experience.

### 3. Build Finite Element Model

The finite element model developed in this paper is based on an axially loaded SRC column made by Wang Ying [27]. The slenderness ratio lo/d = 5, the cross-section is circular, the diameter is 200 mm, and the net height is 1000 mm. The concrete strength grade is C40, the strength grade of the vertical reinforcement of each column is 335 MPa, and the thickness of the concrete protective layer is 25 mm. The finite element model established by ABAQUS software refers to the Chinese standard GB/50010 (2015). The concrete damage plasticity model was selected in ABAQUS for concrete member simulation with Poisson's ratio ν = 0.2. The intrinsic reinforcement model was selected from GB/50010 (2015) with a bilinear model with Poisson's ratio ν = 0.3. Among them, the elastic modulus of vertical reinforcement E1 = 2.0 × 10$^5$ Mpa, the strain-hardening modulus E2 = 0.005 E1, the high strength spiral reinforcement (PC steel bar) modulus of elasticity E1' = 2.06 × 10$^5$ Mpa and reinforcement modulus E2' = 0.005 E1'. The ends of the SRC column finite element model are simplified as end plates with 1000 times the modulus of elasticity of steel and Poisson's ratio ν = 0.1.

The boundary conditions at both ends of the SRC column finite element model: one end is solidified and one end has degrees of freedom in the axial loading direction. The element types of each part of the finite element model: steel tube element type S4R, steel reinforcement element type T3D2, and concrete element type C3D8R, where C3D8R is a reduced order integration element, 4-node general-purpose shell, reduced integration with hourglass control and is a very good choice. It is worth mentioning that reduced integration is a very good strategy [28,29]. The approximate global size of the concrete and reinforcement part of the mesh of the finite element model is 20 mm. It is worth mentioning that the size of the finite element model mesh affects its comparison with the experimental results, but it has less effect on the results of the force state modeling analysis. Figure 2 shows the meshing of each part of the SRC column finite element model and highlights the elements used for modeling correlation as well as stressing state analysis.

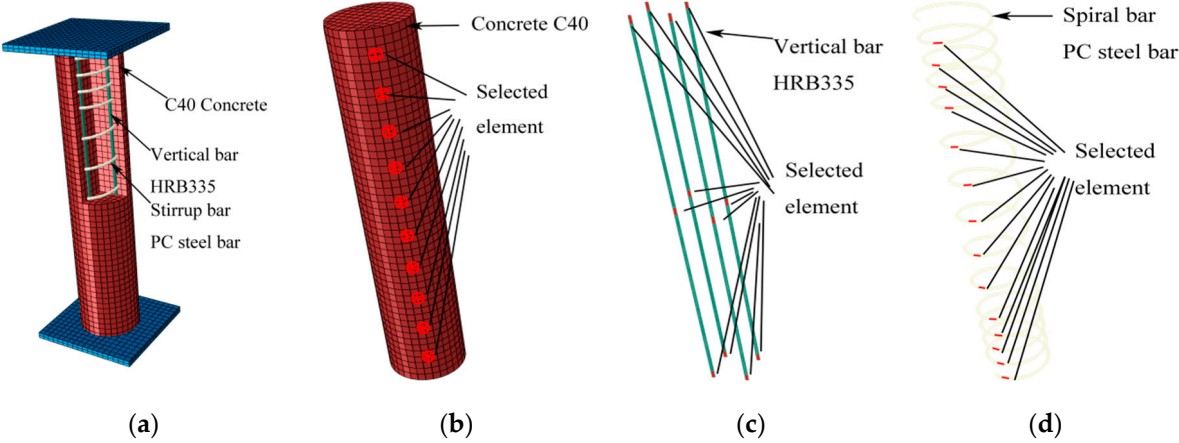

**Figure 2.** Meshing and element selection: (**a**) SRC column; (**b**) Concrete; (**c**) Vertical bar; (**d**) Spiral bar.

The force-displacement curves of the SRC column test and the finite element model are shown in Figure 3. The finite element model established in this paper agrees with the experimental results.

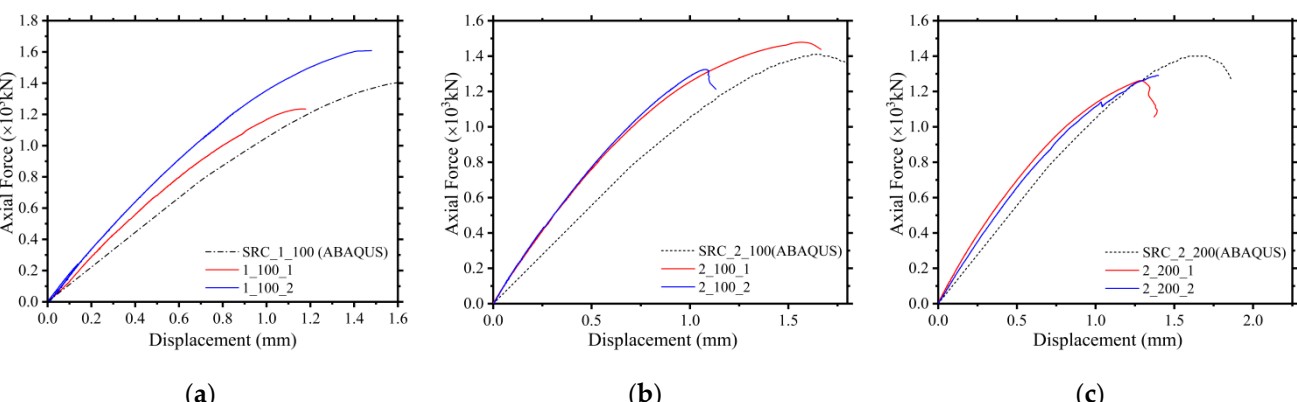

**Figure 3.** Comparison of F-D curves of some test columns and finite element model: (**a**) F-D curve of 1-100; (**b**) F-D curve of 2-100; (**c**) F-D curve of 2-200.

## 4. Stressing State Analysis Based on Correlation Method

### 4.1. Correlation Modeling Analysis of Finite Element Models

By modeling the correlation stressing state of the finite element model SRC_1_100 output element strain energy (*ESE*) as described in Section 2.3, the correlation characteristic parameter curves of concrete, spiral reinforcement, and vertical reinforcement are shown in Figure 4a, and the overall correlation characteristic parameter curve of the SRC column shown in Figure 4b can be obtained. From Figure 4a,b, the mutation law of the evolution of the stressing state of the SRC column can be seen. By applying the slope increment criterion to the characteristic parameter curve of the finite element model SRC_1_100, it can be judged that the load corresponding to its characteristic point P is 733 kN and the ultimate point is 1336 kN.

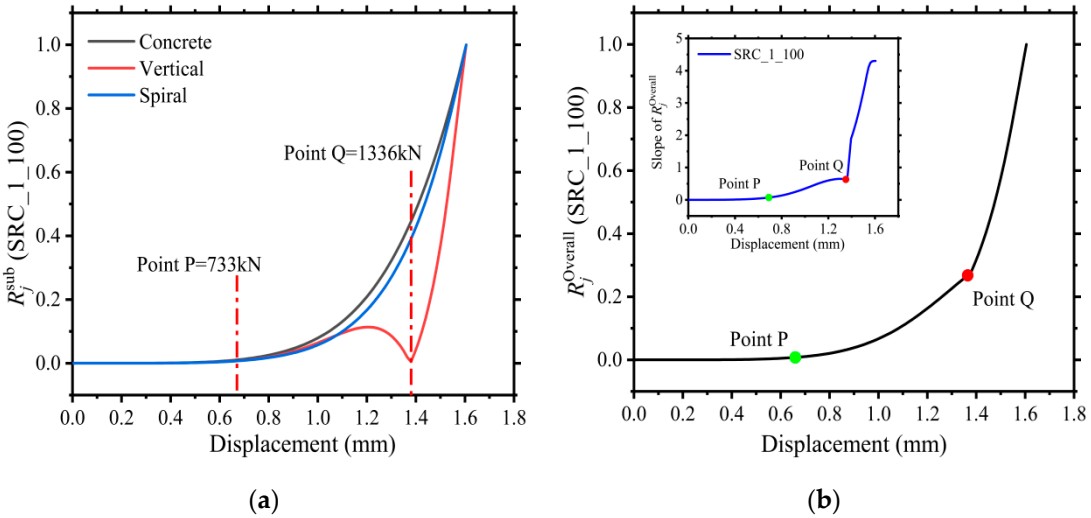

**Figure 4.** FEM-based related characteristic pair of SRC_1_100: (**a**) Sub-mode characteristic parameter curves; (**b**) Overall mode characteristic parameter curve.

The characteristic point P = 733 kN is defined as the failure starting point of the SRC column obtained by modeling the finite element model of the SRC column. The reasonableness of the characteristic point P will be further verified by correlated force state modeling analysis of the test data in the following.

### 4.2. Correlation Modeling Analysis of Experiment

Take SRC columns 1-100-2 and 1-200-2 as examples. The SRC column test strain data were modeled as stressing state modes and characteristic parameters by the correlation modeling analysis method described in Section 2.3. Applying the slope increment criterion to the characteristic parameter curves characterizing the overall stressing state of SRC columns, obtain the characteristic point P of 757 kN for SRC columns 1-100-2 and 727 kN for SRC columns 1-200-2. As seen in Figure 5, the correlation stressing state modes of the two SRC columns also show very obvious mutation near the characteristic point P. The correlation characteristic parameters used to characterize the evolution of the stressing state of the concrete, and spiral reinforcement in the SRC columns also show the same mutation in the turnaround near the characteristic point P.

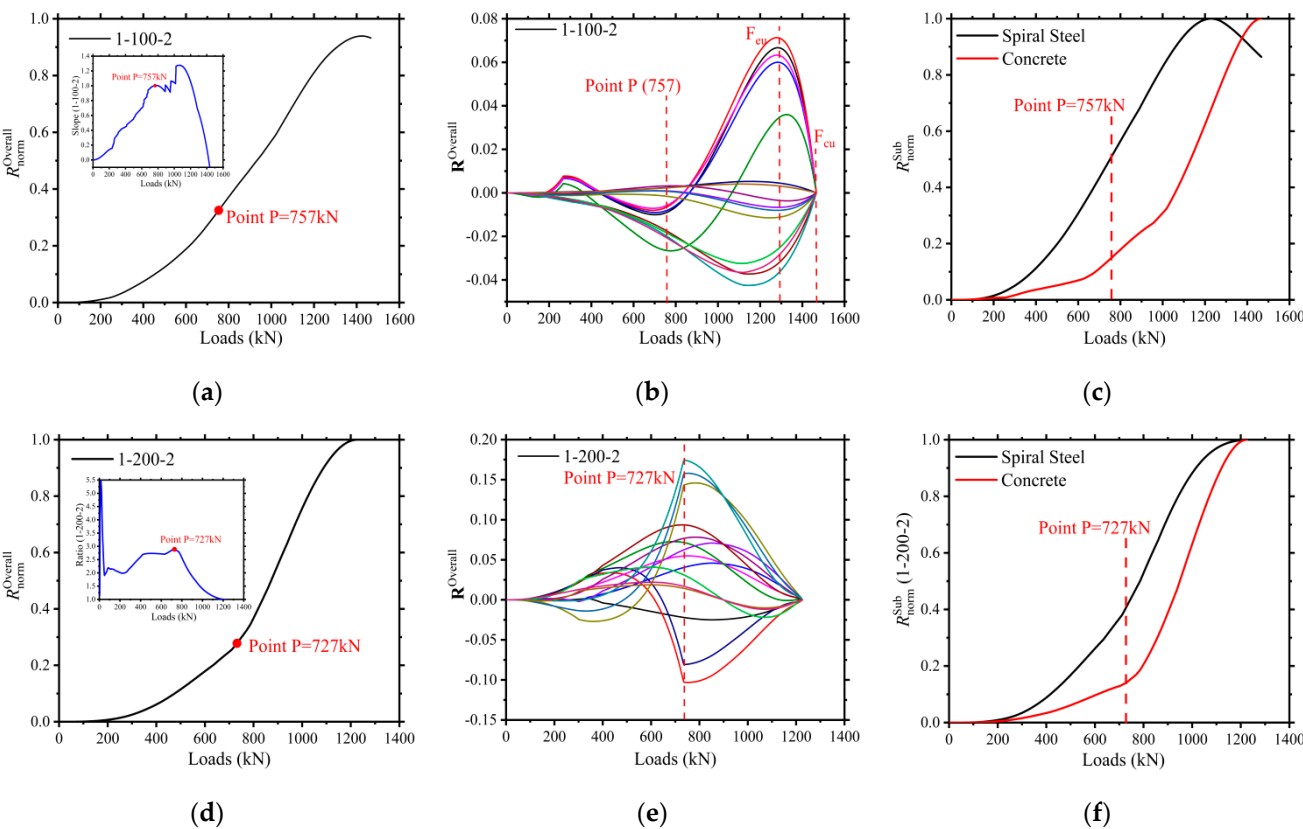

**Figure 5.** Mutation of SRC test's related characteristic pair: (**a**) Correlation parameter of 1-100-2; (**b**) Correlation mode of 1-100-2; (**c**) Correlation parameter of components; (**d**) Correlation parameter of 1-200-2; (**e**) Correlation mode of 1-200-2; (**f**) Correlation parameter of components.

The failure starting point is defined as the characteristic point P obtained by correlation stressing state modeling analysis and slope increment criterion. Taking SRC columns 1-100 as an example, comparing the load values corresponding to the characteristic point P (757 kN) obtained from the modeling test data and the characteristic point P (733 kN) obtained from the modeling finite element model data, there is only a difference of 24 kN in the value corresponding to the characteristic point P. This phenomenon further illustrates the reasonableness of the finite element model established in this paper and shows that both the modeled finite element model and the experimental data can reveal the failure starting point of SRC columns. In addition, the correlation modeling analysis of the SRC column finite element model can also predict the failure starting point of the tested SRC column.

## 5. Mutation Stressing State Characteristic Pair

*5.1. Overall Stressing State Analysis of the Finite Element Model*

In addition to the correlation modeling analysis method, this paper analyzes the internal energy (*IE*) of each part of the SRC column finite element model (concrete, spiral reinforcement, vertical reinforcement) for the overall stressing state; the analysis of the element strain energy (*ESE*) of each part of the finite element model for the partial stressing state can obtain the mutation characteristic presented by the overall and partial SRC column finite element model near the characteristic point P.

As shown in Figure 6, the normalized internal energy (*IE*) of order 0 ($IE_{j,\text{norm}}$), order 1 ($\Delta IE_{j,\text{norm}}$), and order 2 ($\Delta^2 IE_{j,\text{norm}}$) are obtained as three characteristic parameters by modeling the internal energy of the finite element model SRC_100, and the expressions are shown in Equations (3)–(5). Among them, the load corresponding to the characteristic point P is 733 kN, and the load corresponding to the characteristic point Q is 1336 kN. The stressing state characteristic parameters show mutations near the characteristic point, especially for the second-order normalized IE values. The second-order normalized IE value will suddenly drop when the axial load is near the characteristic point P. When the axial load is near the characteristic point Q, the value starts to enter the unstable stage with violent fluctuations. As shown in Figure 6b, the peak load of SRC_1_100 obtained from the finite element model is 1407 kN, which is far from the calculated results based on the Chinese concrete code GB/50010 [5] and the ultimate point load corresponding to the characteristic point P. Therefore, the uncertainty of the peak load of the SRC column is pronounced.

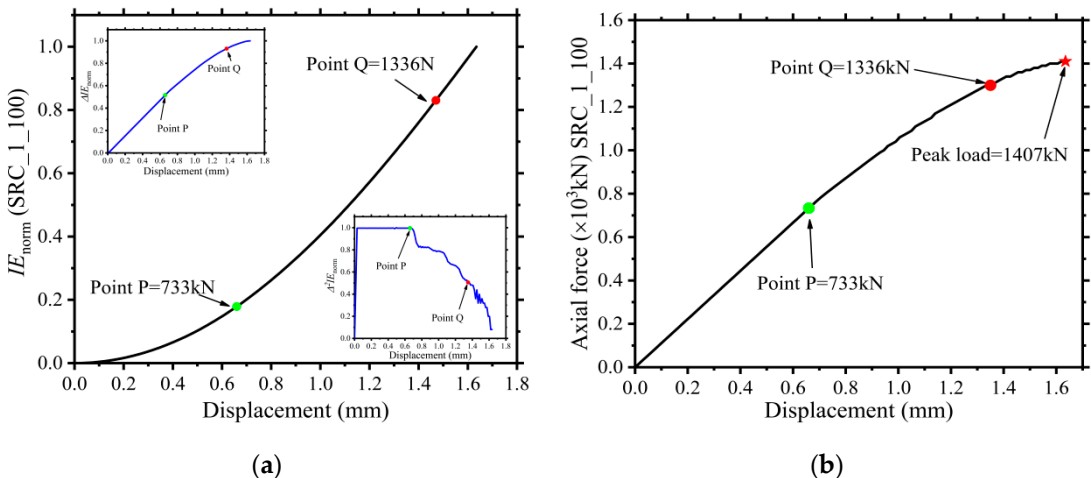

(**a**)                                     (**b**)

**Figure 6.** $IE_{\text{norm}}$-displacement figure and axial force-Displacement figure: (**a**) $IE_{\text{norm}}$-Displacement of SRC_1_100; (**b**) Axial force-Displacement SRC_1_100.

*5.2. Partial Stressing State Analysis of the Finite Element Model*

The selection of the elements used for stressing state modeling analysis for each part of the SRC column has been highlighted in Figure 2. Taking SRC_1_100 as an example, based on the element strain energy (*ESE*) data of the concrete part of the SRC column, three stressing state sub-modes with characteristic parameters have been established, as shown in Figure 7. The sub-modes of concrete stressing states are established in Figure 7a–c and are based on the concrete elemental strain energy *ESE*, the first-order elemental strain energy *ΔESE*, and the second-order elemental strain energy *Δ²ESE*, respectively. Equations (16) and (17) shows the first- and second-order elemental strain energy expressions:

$$\Delta ESE_j = ESE_j - ESE_{j-1} \tag{16}$$

$$\Delta^2 ESE_j = ESE_j - 2ESE_{j-1} + ESE_{j-2} \tag{17}$$

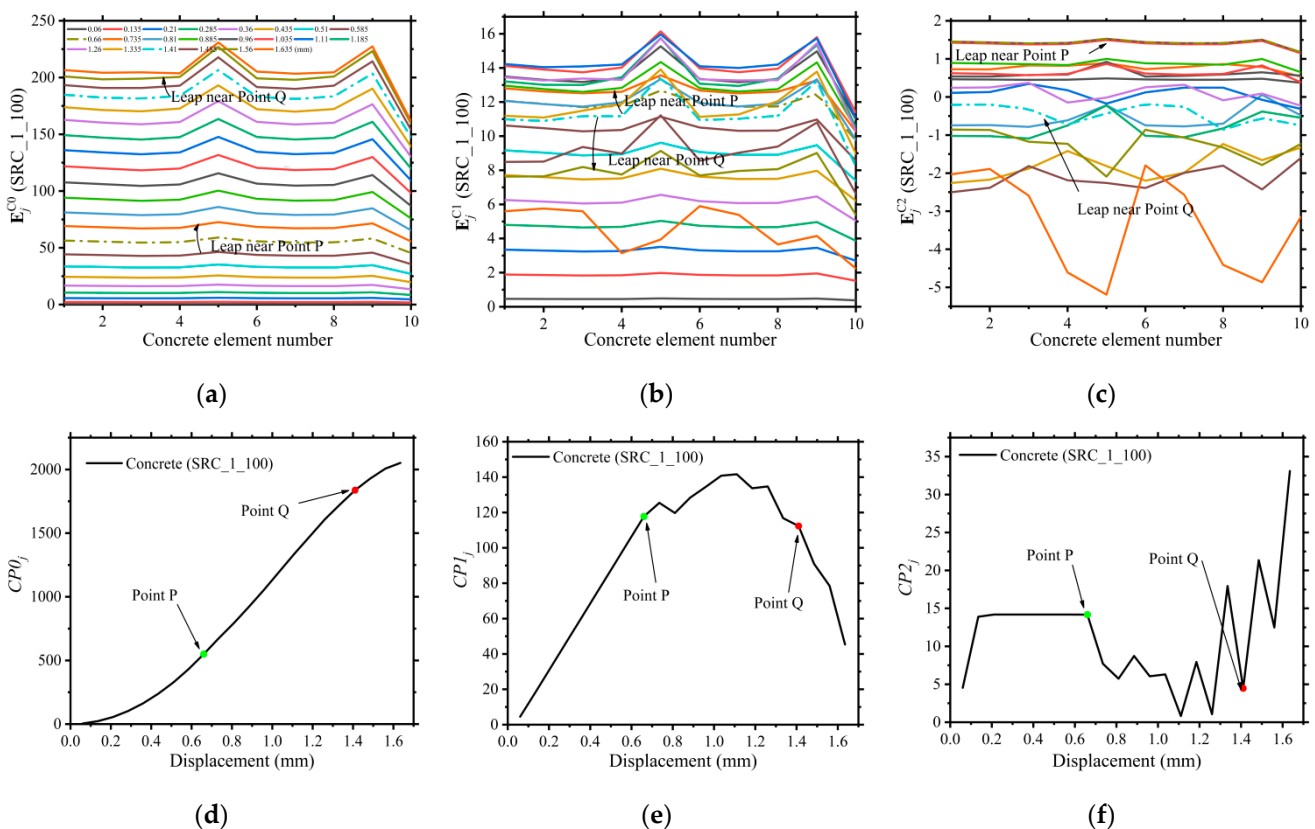

**Figure 7.** Concrete stressing state sub-mode revealed by *ESE*, Δ*ESE* and Δ²*ESE*: (**a**) Mode0 of concrete; (**b**) Mode 1 of concrete; (**c**) Mode 2 of concrete; (**d**) *CP*0 of concrete; (**e**) *CP*1 of concrete; (**f**) *CP*2 of concrete.

The expressions of the numerical modes established based on *ESE*, *DESE*, *D2ESE* are shown in Equations (18)–(20), and the stressing state sub-mode of vertical reinforcement and spiral reinforcement is similar to them:

$$\mathbf{E}_j^{C0} = [ESE_1 \dots ESE_{10}]_j \tag{18}$$

$$\mathbf{E}_j^{C1} = [\Delta ESE_1 \dots \Delta ESE_{10}]_j \tag{19}$$

$$\mathbf{E}_j^{C2} = \left[\Delta^2 ESE_1 \dots \Delta^2 ESE_{10}\right]_j \tag{20}$$

The evolution of the concrete stressing state sub-modes is shown in Figure 7a–c. From the figure, it can be seen that all three concrete stressing state sub-modes show the same mutation near the characteristic point P.

The stressing state sub-mode obtained based on *ESE*, Δ*ESE* and Δ²*ESE* has very obvious mutation near the characteristic point P. Establishing the characteristic parameters corresponding to the stressing state sub-mode can also characterize the evolution of the stressing state of the SRC column. Based on the *ESE*, Δ*ESE*, and Δ²*ESE*, the characteristic parameters *CP*0, *CP*1, and *CP*2 corresponding to the stressing state sub-modes can be constructed, and their expressions are shown in Equation (21):

$$CP0_j = \left|\frac{1}{n}\sum_{k=1}^{n} ESE_{j,k}\right|, \ CP1_j = \left|\frac{1}{n}\sum_{k=1}^{n} \Delta ESE_{j,k}\right|, \ CP2_j = \left|\frac{1}{n}\sum_{k=1}^{n} \Delta^2 ESE_{j,k}\right| \tag{21}$$

Figure 7d–f shows the evolution law of the three characteristic parameters. The positions of the characteristic point P and the characteristic point Q have been marked in the figure. Among them, the change in the characteristic parameter *CP*0 is relatively soft. The

characteristic parameter *CP*1 begins to show a linear change with the loading of the displacement load before the characteristic point P. After the characteristic point P, because the stressing state of the concrete begins to enter an unstable state, the characteristic parameter *CP*1 begins to show a kind of unstable wave change. This feature is more apparent through the characteristic parameter *CP*2 than *CP*1. The evolution of the characteristic parameter *CP*2 is almost a horizontal straight line at the beginning, and after entering the characteristic point P it becomes a sharply fluctuating zigzag-shaped broken line. It can be seen that for the concrete of SRC_1_100, the three characteristic parameters and the three stressing state sub-modes can well characterize the evolution law of the concrete stressing state.

Figure 8 shows the sub-modes with characteristic parameters for the stressing state of vertical reinforcement and spiral reinforcement for SRC_1_100. According to the finite element analysis results, both vertical and spiral reinforcement near the characteristic point P does not start to enter the plastic phase. Near characteristic point Q, the spiral reinforcement enters the plastic phase. However, there is a self-coordinated process in each part of the SRC column during the axial loading process. Therefore, the sub-modes characterizing the SRC column's vertical reinforcement and spiral reinforcement also produce significant mutation near the characteristic point P and characteristic point Q. The corresponding characteristic parameters *CP*1 and *CP*2 also show mutation near the characteristic points P and Q, as shown in Figure 8b,c,e,f.

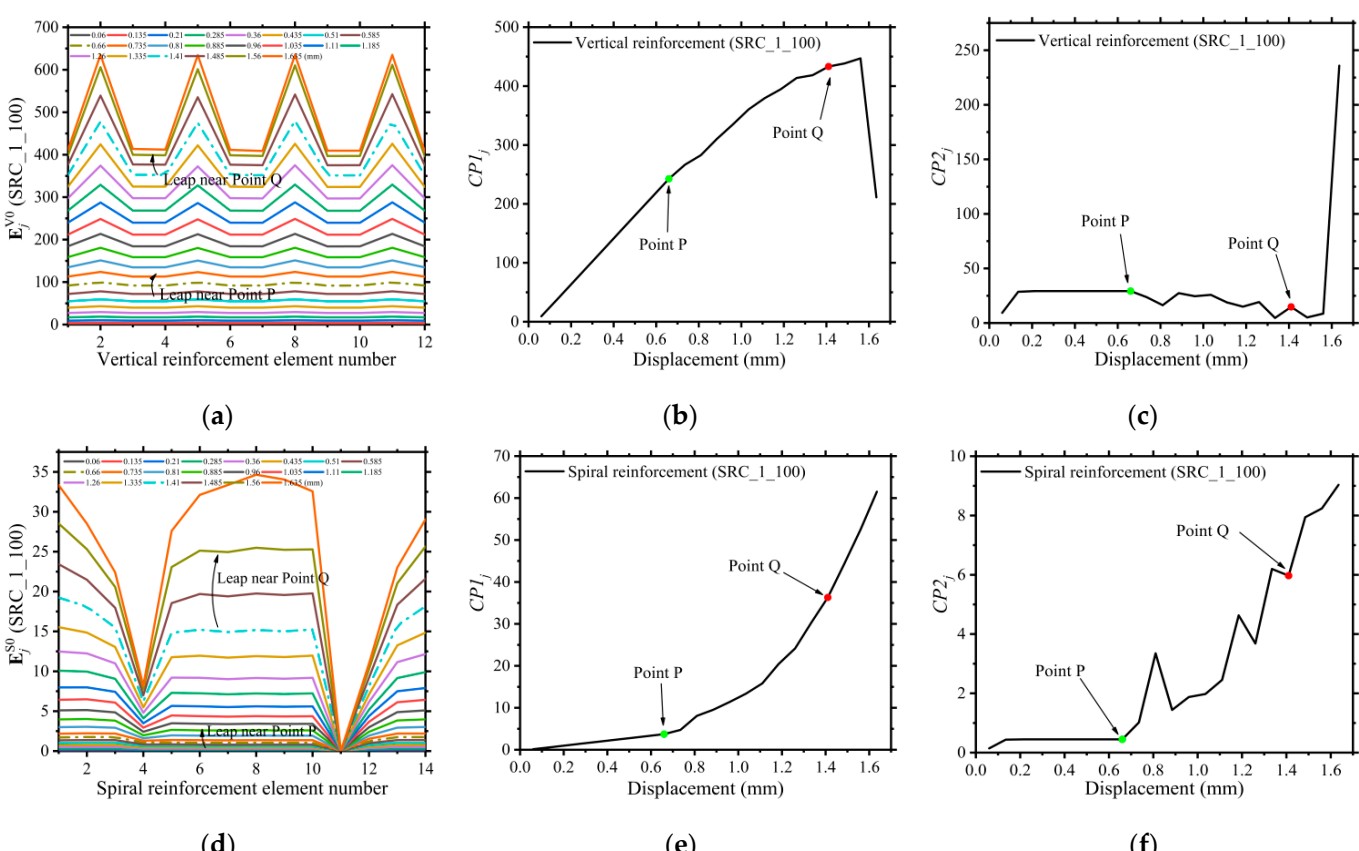

**Figure 8.** Stressing state characteristic pairs of sub-modes based on vertical and spiral reinforcement: (**a**) Mode0 of vertical reinforcement; (**b**) *CP*1 of vertical reinforcement; (**c**) *CP*2 of vertical reinforcement; (**d**) Mode0 of spiral reinforcement; (**e**) *CP*1 of spiral reinforcement; (**f**) *CP*2 of spiral reinforcement.

In the case of SRC_1_100, for example, the failure starting point is 733 kN obtained by modeling the output data of the finite element model, and the failure starting point is 757 kN obtained by modeling the test strain data. The results of the finite element model and test are close. Therefore, we can consider that it is feasible to determine the failure starting point of the SRC column by modeling the finite element model data.

### 5.3. Cloud Diagram Analysis

Figure 9 shows the cloud diagram of the concrete damage plastic compression strain for the SRC_1_100 finite element model near the failure starting point (733 kN). The cloud diagram shows that the finite element model concrete starts to show concrete damage plastic compression when the load step increases to near the failure starting point of 733 kN. It is known that the damage plasticity of concrete is essential for determining the failure starting point of SRC columns, so the cloud diagram analysis further verifies the reasonableness of the modeling method and the characteristic point P of the force state in this paper.

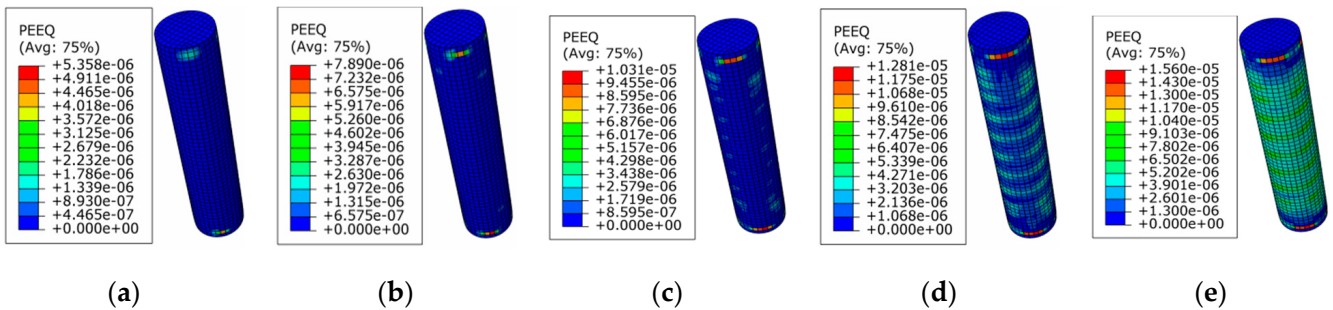

**Figure 9.** Cloud diagram around failure starting point: (**a**) F = 733 kN; (**b**) F = 750 kN; (**c**) F = 766 kN; (**d**) F = 782 kN; (**e**) F = 797 kN.

## 6. Conclusions

In this paper, the feasibility of modeling and analysis of the test and finite element model data of SRC columns by correlation modeling analysis method and the reasonableness of the failure starting point is verified by citing the data from literature [27] and establishing the finite element model of SRC columns. Based on the structural stressing state theory, the correlation modeling analysis is performed on the test strain data of the SRC column and the output element strain energy (*ESE*) data of the finite element model, and the slope increment criterion is applied to reveal the characteristic point P of the SRC column. The reasonableness of the modeling method and the starting point of failure is further verified through the stressing state modeling and cloud diagram analysis of the finite element model output internal energy (*IE*) and element strain energy (*ESE*) data. The main research contents and conclusions of this paper are summarized as follows:

- A finite element model corresponding to the SRC column test in the literature [27] was developed, and the reasonableness of the model was verified;
- A correlation stressing state modeling analysis method applicable to modeling SRC column tests and finite element model data is proposed;
- Based on the correlation characteristic pair, the stressing state modeling analysis, and applying the slope increment criterion, the characteristic points in the failure process of the SRC column are revealed and defined as the starting point of failure.

The failure point obtained from the modeled SRC column test and finite element data are compared, and the cloud diagram characteristics of the SRC column finite element model near the failure point are observed to verify the feasibility of predicting the failure point of the test SRC column by the finite element model.

**Author Contributions:** Conceptualization, Z.S. and B.L. methodology, Z.S.; formal analysis, Z.S.; investigation, Z.S.; resources, Z.S.; writing—original draft preparation, Z.S.; writing—review and editing, B.L.; visualization, Z.S.; supervision, G.Z. All authors have read and agreed to the published version of the manuscript.

**Funding:** This research received no external funding.

**Institutional Review Board Statement:** Not applicable.

**Informed Consent Statement:** Not applicable.

**Data Availability Statement:** The data presented in this study are available on request from the corresponding author.

**Conflicts of Interest:** The authors declare no conflict of interest.

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
