# Peer review of "Stressing State Analysis of SRC Column with Modeling Test and Finite Element Model Data"

_applsci, doi:10.3390/app12178866_

Round 1
Reviewer 1 Report
This paper contains valuable contents and I recommend its publishing, as a minor point a short description about mesh size and its effect on the results could be put in the text.
Reviewer 2 Report
Dear authors of the manuscript entitled “General failure rule of SRC column revealed by the structural stressing state theory”,
I have carefully reviewed your manuscript. The topic seems fit with the scope of the journal. However, you are required to consider the following issues before resubmitting the manuscript for publication:
1- Please rewrite the title according to the journal style
2- There are some typing and grammar mistakes. Therefore, I suggest reviewing the entire manuscript by a professional editing service
3- The abstract section should be one paragraph up to (250-300) words, summarizes the major aspects of the entire paper in a prescribed sequence that includes: 1) the research problem; 2) the overall purpose of the research; 3) the methodology and/or data used; 4) major findings or the important outcomes of the research. Therefore, the research problem is missing, as it should be mentioned at the beginning of abstract. Hence, please reconstruct the abstract accordingly. Moreover, please avoid the citation in the abstract.
4- Some symbols need to be defined in the abbreviations paragraph. Please consider revision
5- The references in table 1 and table 2 are not cited. Please revise
6- References are not all in the same format. Please use a proper tool for citation and follow the journal style
7- Please download the citation from original source (e.g., Science direct, Springer, …etc.). not simply use google scholar or other databases.
8- Finally, I have not seen any connection between your research and articles already published in this journal. Please consider this matter
Therefore, the reviewer believes that the manuscript is not suitable for publication in its present form due to the above reasons
Reviewer 3 Report
The main topic of the paper is good but the technical presentation of the methodology and related results is definitively poor (it can be strongly improved). The idea behind this research is sound and the technical content of the paper is considered to be enough. Nonetheless, this reviewer would like to bring many major comments/suggestions (and also curiosities) to the attention of the authors, which will hopefully increase the quality of the manuscript.
Some suggestion not in order of importance (to be addressed):
- One general comment: English should be greatly improved! I suggest to the authors to, more or less, re-write many parts of the paper to make it more readable and attractive (even more understandable). Some examples taken from the “Abstract”:
- In the abstract “…..the word “data” is repeated seven time in the first 8 lines of the abstract;
- Many sentences in the abstract are very long;
- Please remove acronyms from the abstract FEM, SRC, CP…they should be used for the first time and defined in the main body of the paper.
Now, I move to more technical issues:
- Authors say in the “Introduction” to the paper “SRC columns are widely used in engineering due to their high bearing capacity, deformation capacity, and energy consumption capacity. Since the beginning of the twentieth century, many scholars worldwide have studied the bearing capacity of SRC columns from the perspective of constitutive models of confined concrete and concrete strength criteria.” This could be true but some references are needed. Please look in scopus or google scholar and try references (there are many of them) to motivate these sentences.
- In Table 1 many double squares appear “[]” without references. Furthermore in Table 1 appears “Error! Reference source not found.”
- Authors say “Most of the world's concrete structure codes”. This again as above could be true but please introduce some codes references. Eurocode ? ACI ? or others.
- Reviewer thinks that all the references are missing or wrong. Here is what appear when I read “For example, Nagashima5, Issa MA6, Suzuki M7, Akiyama M8, Wu T9, and others proposed various…” For sure it is an error !! In page 4 line 118 again it appears “shellError! Reference source not found”.
- Paragraph 2.2 has this title “Stressing state characteristic pair based on IE”. Again please avoid acronyms from the title and make title more understandable and readable.
- Author simply introduce in one line “C3D8R”. It is a reduced order integration element. 4-node general-purpose shell, reduced integration with hourglass control is a very good choice. It is a reduced integration FEM. Very detailed FE model !! Probably what I’m suggesting to the authors is to spend a couple of sentences more on this topic. The research is treating a very important aspect of this kind of structures and the reviewer is deeply impressed by the amount of numerical research done. However a few lines have been dedicated to a very complex FE model which seems to the reviewer to consistently “fit” the real behavior. So, in the opinion of the reviewer, I suggests that major information should be added along this line:
- number of elements and degrees of freedom of the full model and number of gauss (or lobatto) points used during the integration; integration scheme adopted; more information on the iteration’s algorithm (Newton, Newton-Raphson ordinary or modified, arc-length, etc), etc. (not all the information are required if they are outside the scope of the research or simply not used);
- which kind of nonlinearities used: geometrical, material, or both; large or small strains used; etc.
Furthermore “reduced integration” is a very good strategies. I suggest to add a couple of references to help the reader to understand this concept: a good paper is https://doi.org/10.1515/cls-2022-0027 and T. Belytschko, B. L. Wong and H-Y. Chiang, (1992) Advances in one-point quadrature shell elements, Computer Methods in Applied Mechanics and Engineering, 96(1), 93-107.
Round 2
Reviewer 2 Report
Dear authors of the manuscript entitled “General failure rule of SRC column revealed by the structural stressing state theory”,
I have carefully reviewed your revised manuscript and have concluded the following remarks:
1- The manuscript is generally well revised with a good reconstructed abstract, comprehensive introduction, and an intense discussion of the results, including tables, figures, diagrams, …etc.
2- Author responses to reviewers’ comments are well addressed.
3- Further language check is required.
4- If the authors believe that many papers published in this journal are related to their research, why have not seen any mention of such references?
Reviewer 3 Report
Authors properly answered all the reviewer's comments. Good job